# TableTextGrad: A Reflexive Framework for Table Understanding

## Abstract

Table understanding is a complex task that requires not only grasping the semantics of free-form questions but also accurately reasoning over semi-structured tables. Recently, promising approaches designed sophisticated prompts that leverage large language models (LLMs) by combining Chain-of-Thought strategies with function calls, consequently demonstrating competitive results without requiring fine-tuning. However, creating sufficiently effective prompts remains a challenge. Without fine-tuning, all necessary priors must be incorporated directly into the initial prompt, making prompt design even more critical. Motivated by the recent advancements in the "textual gradient" space, we introduce TableTextGrad, a novel framework that enables automatic prompt optimization by leveraging the "differentiation" of prompting pipelines through textual gradients. Concretely, according to the feedback of LLMs, TableTextGrad iteratively refines each function within the Chain-of-Thought steps and function calls, resulting in more accurate and reliable table reasoning outcomes. Experiments on table question-answering datasets demonstrate that our integrated approach achieves significant improvements, setting a new state-of-the-art results on WikiTableQA. Our TableTextGrad not only enhances the reasoning capabilities of LLMs in the table reasoning task but also lays a groundwork for more robust and generalizable prompting pipelines due to its simplicity and effectiveness.

## 1 Introduction

Table understanding and reasoning are crucial in business and consumer applications (Cafarella et al., 2008), as tables typically contain well-structured data that can be efficiently queried using SQL or Python. However, reasoning over tables remains challenging due to factors such as ambiguous feature names and complex relationships between columns, which hinder precise information retrieval from the table as well as accurate query interpretation and reasoning. Recent advances in large language models (LLMs) have demonstrated potential in overcoming these challenges, particularly in tasks like fact verification (Chen et al., 2019) and question answering (Jin et al., 2022; Pasupat & Liang, 2015; Nan et al., 2022).

Approaches for LLM-based table reasoning can be broadly divided into two categories. The first involves fine-tuning models by adjusting LLM embeddings, attention mechanisms (Herzig et al., 2020; Wang et al., 2021; Gu et al., 2022), or training models to improve SQL generation directly (Eisenschlos et al., 2020; Liu et al., 2021; Jiang et al., 2022). The second category leverages inference-only techniques like Chain-of-Thought (CoT) reasoning and in-context learning (ICL) (Chen, 2023; Cheng et al., 2022; Ye et al., 2023; Hsieh et al., 2023; Liu et al., 2023; Wang et al., 2024) to boost performance without fine-tuning.

Each approach has its drawbacks: fine-tuning is computationally intensive and lacks flexibility for new tasks due to its reliance on task-specific labeled data. In contrast, inference-only table understanding offers adaptability but fails to fully utilize labeled data. Recent research has shown the potential of leveraging labeled data for prompting methods (Singh et al., 2023; Gulcehre et al., 2023; Agarwal et al., 2024; Yuksekgonul et al., 2024) to boost LLM performance without the need for fine-tuning. Notably, TextGrad, a recently introduced framework, performs automatic "differentiation" through text, using natural language feedback from LLMs to optimize their outputs. In our case, we may apply TextGrad to refine prompt optimization for table understanding.

Our proposed TableTextGrad extends TextGrad's capabilities by dynamically adjusting prompts in multiple chain-of-thought steps and multiple branching function calls, combining the strengths of inference-only flexibility with data-driven learning to improve table understanding tasks. Additionally, we perform experiments on non-destructive functions that perform soft selection (*italicizing* relevant cells) rather than hard selection, which may remove relevant information (Patnaik et al., 2024). Through a training process, TableTextGrad advances LLM capabilities in handling complex table-based tasks, achieving state-of-the-art results in TabFact and WikiTableQA.

Figure 1: This figure presents TableTextGrad, which refines prompts through natural language feedback and gradient updates on training data. We demonstrate how prompts are iteratively improved through text gradients. The training, validation, and testing phases are similar to the general ML training pipeline. The best-performing prompt on validation is then saved. Chain of Table Inference (in blue) is the chain-of-thought table understanding pipeline that utilizes prompt-based operations for table inference using a set of actions (e.g., add column, filter rows). The table is updated after each step. The Gradient Update (in green) is the textual gradient used to refine the table understanding prompts.

We summarize our contribution as follows:

- We present TableTextGrad, an advanced extension of the TextGrad framework, designed to dynamically optimize prompts in multi-step reasoning tasks. By incorporating multiple chain-of-thought steps and branching function calls, TableTextGrad effectively combines the adaptability of inference-only techniques with the robustness of data-driven learning, improving LLM performance in table understanding tasks.

- Our approach introduces non-destructive functions that perform soft selection of table elements (e.g., *italicizing* relevant cells) instead of hard selection, which risk excluding critical information. This ensures a more nuanced understanding of the tabular data without removing potentially useful context, enhancing overall table comprehension.

- Through extensive experiments, TableTextGrad achieves new state-of-the-art (SOTA) results on key benchmarks like TabFact and WikiTableQA, significantly improving LLM accuracy and reasoning in complex table-based queries.

## 2 RELATED WORK

### 2.1 TABLE UNDERSTANDING

Recent advancements in machine learning and data processing have led to innovative solutions for table-related QA. Large, pretrained LLM on multiple tables (Zhang et al., 2023; Li et al., 2023; Jiang et al., 2022; Xie et al., 2022) propose versatile LLMs trained to perform a variety of tasks such as reasoning, completion, QA, and more (Zha et al., 2023; Yang et al., 2023). Finetuned LLMs are surprisingly good in this space, with subtable selection and reasoning improvements (Zhao et al., 2022; Gu et al., 2022; Patnaik et al., 2024). Similarly, LLM Prompting has seen success due to LLM's powerful inherent reasoning abilities (Cheng et al., 2022; Ye et al., 2023; Jiang et al., 2023; Wang et al., 2024).

### 2.2 LLM PROMPTING FOR TABULAR UNDESTANDING

The are multiple widely used strategies to provide models with instructions for improving downstream tasks to prompt LLMs. Chain-of-Thought (CoT) (Wei et al., 2022) suggests generating reasoning steps before producing an answer rather than directly generating an end-to-end solution. Building on CoT, Least-to-Most (Zhou et al., 2022) and DecomP (Khot et al., 2022) break questions into subproblems, where each step builds on previous ones. This task decomposition improves performance on complex problems by using intermediate subproblem results. Jin & Lu (2023) extends CoT with a table-filling approach, mainly for text-based tasks. As Chen (2023) reports, generic reasoning methods work reasonably well with LLMs, but there are gaps compared to table-specific methods (Cheng et al., 2022; Ye et al., 2023).

Still, CoT-based methods tailored to tabular data generally utilize external tools. Chen et al. (2022); Gao et al. (2023) suggest using Python programs to solve reasoning tasks, significantly improving arithmetic reasoning. Text-to-SQL (Rajkumar et al., 2022) applies this approach to table understanding, while Binder (Cheng et al., 2022) generates SQL or Python programs and extends their capability by calling LLMs as APIs. LEVER (Ni et al., 2023) further verifies the generated programs through execution results. However, these program-aided methods struggle with complex tables due to limitations of *single-pass* generation, where LLMs cannot dynamically modify tables based on specific questions, relying instead on static tables. In contrast, our method adopts a *multi-step* reasoning framework that iteratively transforms tables to suit the given question.

Dater (Ye et al., 2023) and Chain of Table (Wang et al., 2024) modify tabular context during reasoning. Dater was the first to introduce table decomposition, but mainly focused on data pre-processing, with operations limited to fixed column and row selections. Chain of Table generalized a wider range of table operations and *dynamically* generates reasoning chains based on input, leveraging LLMs' planning capabilities (Valmeekam et al., 2022; Hao et al., 2023). Despite these advancements, both approaches rely on quality, human-expert annotated initial prompts, with no easy way to tune prompts beyond manual trial and error.

### 2.3 AUTOMATED LLM CORRECTION:

The idea of correction in LLM Agents has been recently popular (Agarwal et al., 2024; Singh et al., 2023; Gulcehre et al., 2023; Shinn et al., 2024; Huang et al., 2023; Feng et al., 2024; Yuksekgonul et al., 2024). The concept of "Reinforced ICL" (Agarwal et al., 2024) evaluates the CoT rationals on labeled data and retrieves reference data in the test time. While effective, this work does not explore the idea of error case correction or adding additional Prompt Conditions. Similarly, "prethinking" on an unlabeled dataset, saving the high-confidence thoughts, and retrieving them boosts performance at inference-time for QA tasks (Li & Qiu, 2023). Huang et al. demonstrated that self-correction without ground truth does not perform well (Huang et al., 2023), which we also observed. Corrective retrieval has also been proposed (Yan et al., 2024; Asai et al., 2023)–Asai et al. demonstrated that finetuning an LLM to learn to retrieve raw data is beneficial for QA and long-form generation (Asai et al., 2023). Self-correction of text-to-SQL using ICL (Pourreza & Rafiei, 2024) has also been explored. However, to our knowledge, no approach has focused on the automatic correction of prompts for table understanding like in TableTextGrad.

## 3 METHODOLOGY

The general process is shown in Figure 1. The TableTextGrad framework is designed for automatic prompt updating, enabling large language models (LLMs) to refine their reasoning over tabular data through an iterative process that combines natural language feedback and gradient updates. We first describe the general table understanding framework.

### 3.1 CHAIN OF TABLE BACKBONE

For general table understanding, we use Chain of Table (Wang et al., 2024) as the backbone, where LLM Agents engage in step-by-step, function-aided reasoning over the Table and Question, shown in Algorithm 1. We briefly overview how inference works in this section.

We convert the tables into a list of strings to make the tables interpretable by LLMs. For a given table-based reasoning task, we represent the given paired (table, query) as $(T, Q)$, where $T$ stands for the table and $Q$ represents a table-based question or a statement to be verified (to accommodate TabFact). The objective of the LLM is to predict the answer based on the corresponding $(T, Q)$.

---

**Algorithm 1** TableTextGrad Chain of Table Backbone

**Inputs:** Table $T$ and Question $Q$.
**Outputs:** $\hat{A}$ predicted answer.
1: $chain \leftarrow []$
2: **while** $f \neq$ END **do**
3:    $f \leftarrow$ prompt_next_function$(T, Q, chain)$       # Get next table function
4:    $args \leftarrow$ prompt_f_args$(T, Q, f)$       # Get arguments specific to table function $f$
5:    $T \leftarrow f(args, T)$       # Apply processing to Table $T$
6:    $chain \leftarrow chain + [f, args]$       # Update the chain of thought
7: **return** prompt_final_query$(T, Q)$       # The output is predicted answer $\hat{A}$

---

The set of all functions $f$ is described as follows:

- `add_column` adds an additional column that may contain intermediate calculations. For example, if a table about athletes has `Jennifer (US)`, `Josh (UK)`, the model could call `add_column(country, [US, UK])`.

- `group_by` returns a secondary table (appended to the original table) of the count of each unique element in a column. This is similar to the pandas `value_counts` function.

- `select_row` retains only certain relevant rows in the table.

- `select_column` retains only certain relevant columns in the table.

- `sort_by` sorts a column based on its numerical values, and the order can be specified (small-to-large or the reverse).

Note that by default, Chain of Table's `select_row` and `select_column` *remove* information from the table (*hard selection*). However, in our proposed *soft selection*, we simply *italicize* the intersection of selected rows and columns, as shown in Figure 2. In raw text prompt format, we do this by adding asterisks to any *italicized text*. prompt_next_function is a prompt that generates one of the functions $f$ t. At any point, if no further processing is needed an END tag is predicted. prompt_f_args is a separate prompt that generates the arguments to the specific $f$. The separate nature of this allows many ICL examples of function $f$ usage to be shown, improving performance. Finally, prompt_final_query is the final prompt that asks the LLM to predict the answer after all $f$ table processing. For all prompts, multiple ICL examples are also included.

### 3.2 TABLETEXTGRAD

We overview our main contribution. TableTextGrad works similarly to the standard Machine Learning training pipeline. First, an initial LLM (Agent 1) uses the Chain of Table backbone to iteratively generate table operations for table understanding, such as adding columns or filtering rows. After each step, the table is updated based on the generated function calls and function arguments, allowing for incremental selection and processing of relevant table data.

Next, in the Validation Phase, a second LLM agent (Agent 2) evaluates the predicted answers from Agent 1 for the table QA task using text matching (after processing to remove formatting). Natural language feedback of how to improve the prompt given any incorrect predictions is then backpropagated as textual gradients, which are backpropagated to every prompting step used to generate the answer, encompassing all prompts used for function selection, function argument generation, and final table query. This refined prompt is validated by rerunning the new prompts for the Chain of Table on a validation set, and saved if the performance is better than the current set of prompts.

Finally, after a certain number of batches, the best-performing set of prompts is returned. We detail TableTextGrad more formally in Algorithm 2.

---

**Algorithm 2** TableTextGrad Table Understanding

---

**Inputs:** $\mathcal{D}_{train}, \mathcal{D}_{valid}, \mathcal{D}_{test}, P_{init}$ is the training, validation, and test splits, and $P_{init}$ is the initial prompt. Each $\mathcal{D}$ is a set of Tables $T$ and Questions $Q$.

**Outputs:** $P_{tuned}$ is the tuned version from the initial prompt.

1: $P_{tuned} \leftarrow P_{init}$
2: # Obtain current Chain of Table inference performance on validation data for comparison
3: $loss_{val} \leftarrow \sum loss\_fn(\text{COT}(T, Q, P_{init}), A), \ \forall \, T, Q, A \in \mathcal{D}_{valid}$
4: **for** Batch $\in \mathcal{D}_{train}$ **do**
5:     $loss \leftarrow 0$
6:     **for** $T, Q, A \in$ Batch **do**
7:         $\hat{A} \leftarrow \text{COT}(T, Q, P_{tuned})$               # Chain of Table onference
8:         $loss$ += loss\_fn$(\hat{A}, A)$              # String matching boolean for Table QA
9:     $loss$.backward()                  # Backpropagate textual gradients from $loss$
10:     $P^* \leftarrow$ optimizer.step()           # Obtain potentially better performing prompts
11:     $loss^*_{val} \leftarrow \sum loss\_fn(\text{COT}(T, Q, P^*), A), \ \forall \, T, Q, A \in \mathcal{D}_{valid}$
12:     **if** $loss^*_{val} < loss_{val}$ **then**
13:         $loss_{val}, P_{tuned} \leftarrow loss^*_{val}, P^*$         # Save better performing prompts and $loss_{val}$
14: **return** $P_{tuned}$

---

The ".backward()" call is an LLM prompt that asks Agent 2 for criticisms to improve $P_{tuned}$ given $loss$. This call is repeated to other parameters in the gradient graph using backpropagation. I.e. if the call was $X \rightarrow Y \rightarrow loss$, the gradient backpropagation would look like the outputs to the following prompt:[1]

> $\frac{\partial loss}{\partial X}$ = Here is a conversation $X, Y$. Here are the criticisms on $Y$: $\frac{\partial loss}{\partial Y}$. Give some criticisms on improving $X$.

Similarly, the optimizer.step() call is an LLM prompt that asks Agent 2 to return an updated $P^*$ that incorporates the criticisms from the backward call. We note that there is no learning rate, and the optimizer.step() function is a prompt to Agent 2 on how the current parameters can be improved based on $loss$. Additionally, while Agent 1 and Agent 2 may be the same LLM, in practice, we use more powerful models for Agent 2 vs Agent 1 in order to have better possible textual gradients.

Furthermore, because we rely on LLM output, $loss$.backward() and optimizer.step() prompts may crash due to length constraints / general power of the LLM. To reduce this risk, we found that explicitly excluding lengthy ICL examples from $P_{init}$ and adding that as a prompt input (i.e. adding ICL examples to $Q$ instead) was useful.

## 3.3 DATASETS AND BASELINES

We assess TableTextGrad on two commonly used datasets: WikiTableQA (WikiTQ) (Pasupat & Liang, 2015) and TabFact (Chen et al., 2019) (Table 1). WikiTQ focuses on table-based question answering, demanding complex reasoning over tables with short-text answers, whereas TabFact is a benchmark for binary fact verification, evaluating the truthfulness of state-

Table 1: Dataset Statistics

| | WikiTQ | | TabFact | |
|---|---|---|---|---|
| | Questions | Tables | Questions | Tables |
| Train | 14,148 | 1,679 | 92,283 | 13,182 |
| Valid | 3,536 | 1,455 | 12,792 | 1,696 |
| Test | 4,344 | 421 | 2,024 | 298 |

---

[1]This example is taken directly from Yuksekgonul et al. (2024)

ments derived from table data. Consistent with prior research, we report performance metrics using cleaned string matching for WikiTQ and binary prediction accuracy for TabFact.

Both WikiTQ and FeTaQA are datasets aimed at table-based question answering, requiring sophisticated reasoning across tables. WikiTQ typically involves short-text span answers, while FeTaQA asks for more detailed, free-form responses. Conversely, TabFact is a binary fact verification task that requires determining whether a given statement is true or false based on table data. For WikiTQ, we evaluate performance using string matching accuracy (post-processing for consistency), and for TabFact, we use binary classification accuracy as the metric.

The baseline methods are divided into two categories. Finetuning-based are methods that require training the weights of a base model. This includes methods like Unifiedskg (Xie et al., 2022), PASTA (Gu et al., 2022), and CABINET (Patnaik et al., 2024).

The second category are inference only methods such as Chain-of-Thought (Wei et al., 2022), Text-to-SQL (Rajkumar et al., 2022), Binder (Cheng et al., 2022), and Dater (Ye et al., 2023). Chain-of-Thought (Wei et al., 2022) prompts the LLM to explain its reasoning process before answering the question. Text-to-SQL (Rajkumar et al., 2022) uses in-context examples to guide the LLM in generating SQL queries for answering questions (Chen et al., 2022; Gao et al., 2023). Binder (Cheng et al., 2022) combines a language model API with SQL or Python to generate executable programs that reason over the table. Dater (Ye et al., 2023) uses few-shot examples to decompose complex table contexts and questions into smaller sub-tables and sub-questions, enhancing table reasoning.

Table 2: Accuracy comparisons of all baselines vs Table-TextGrad. Results are copied from the original papers' most relevant and best-performing configurations (missing results are denoted with a dash "-"). The best performance is **bolded**. The second best performance is underlined. Chain of Table$^*$ denotes our backbone implementation in TableTextGrad, without any tuning. TableTextGrad$_{SF}$ denotes our method with soft selection and full pipeline gradient tuning.

| Approach | Base Model | TabFact | WikiTQ |
|---|---|---|---|
| **Finetuning-Based** | | | |
| Unifiedskg (Xie et al., 2022) | T5 3B | 83.68 | 49.29 |
| REASTAP (Zhao et al., 2022) | BART-Large | 80.1 | 58.6 |
| PASTA (Gu et al., 2022) | DeBERTaV3 | 85.60 | - |
| OmniTab (Jiang et al., 2022) | BART-Large | - | 62.80 |
| CABINET (Patnaik et al., 2024) | BART-Large | - | 69.10 |
| **LLM Prompting** | | | |
| BINDER (Cheng et al., 2022) | GPT-3 Codex | 86.00 | 64.60 |
| DATER (Ye et al., 2023) | GPT-3 Codex | 85.60 | 65.90 |
| STRUCTGPT (Jiang et al., 2023) | GPT 3.5 | 87.60 | 57.00 |
| Chain-of-Thought (Wei et al., 2022) | PaLM 2 | 79.05 | 60.43 |
| E5 (Zhang et al., 2024) | GPT-4 | **88.77** | 65.54 |
| Chain of Table (Wang et al., 2024) | GPT 3.5 | 80.20 | 59.94 |
| Chain of Table (Wang et al., 2024) | PaLM 2 | 86.11 | 67.31 |
| Chain of Table$^*$ | Llama 3.1 70B | 85.05 | 63.58 |
| Chain of Table$^*$ | GPT 4o mini | 81.20 | 60.34 |
| Chain of Table$^*$ | GPT 4o | 86.41 | 64.95 |
| TableTextGrad$_{SA}$ | Llama 3.1 70B | 87.05 | 70.58 |
| TableTextGrad$_{SA}$ | GPT 4o mini | 86.62 | 64.14 |
| TableTextGrad$_{SA}$ | GPT 4o | 88.75 | **75.10** |

Note that we slightly distinguish between the default Chain of Table implementation and our reimplementation with a $^*$, since small changes may slightly affect downstream performance.

## 4 RESULTS

We see the results in Table 2 to the right (FeTaQA results are shown in Appendix A.3). The table presents accuracy comparisons across different approaches for the TabFact and WikiTQ datasets, In finetuning-based methods, which involve model adaptation to specific tasks, PASTA (Gu et al., 2022) performs well on TabFact with accuracies of 85.60, while CABINET (Patnaik et al., 2024) leads WikiTQ with 69.10%. However, these methods require extensive finetuning on the dataset, which can limit generalizability.

While fine-tuning methods can provide high accuracy, the versatility and competitive performance of LLM prompting strategies also offer compelling performance. Models leverage pre-trained LLMs without task-specific finetuning, both the Chain of Table base model and our re-implementation demonstrate strong baseline performance, achieving competitive results. The GPT 4o version

achieves the best performance on both TabFact (86.41%) and WikiTQ (64.95%) out of the box, surpassing prior approaches by using more recent LLMs.

The proposed TableTextGrad approach (highlighted as TableTextGrad and TableTextGrad $_SA$) demonstrates impressive results in this LLM prompting setting. Notably, TableTextGrad $_SA$, which incorporates soft selection and the full pipeline gradient tuning, achieves strong performance with 88.75% on TabFact (within .02 from SOTA) and 75.10% on WikiTQ, highlighting the Table-TextGrad's effectiveness. These results show that gradient-based refinement techniques help optimize task-specific accuracy, with little to no human effort.

## 4.1 Soft vs Hard Table Selection

Figure 2 illustrates the difference between hard and soft table selection.

### Hard vs Soft Table Selection

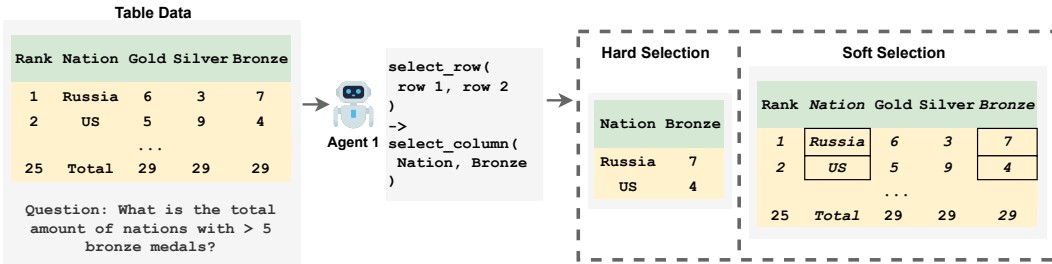

Figure 2: On the left, a table with data on nations' medal counts is presented, along with a question about the total number of nations with more than 5 bronze medals. In the center, an agent performs a hard selection by choosing specific rows and columns, reducing the table to only the relevant data (Russia and US in the "Nation" and "Bronze" columns). On the right, the soft selection highlights (in italics) the relevant cells without excluding the rest of the table's content. This approach retains broader contextual information, allowing for a more comprehensive understanding of the data while emphasizing critical details.

Table 3: Ablations: Table understanding results on WikiTQ and TabFact with GPT 4o mini, GPT 4o, and Llama 3.1 70b. $H$ and $S$ denotes hard and soft selection respectively. $A$ and $L$ denote all prompts tuned vs only the last prompt tuned respectively (underline denotes the second-best performance; **bold** denotes the best performance)

| | Llama 3.1 70B | | GPT 4o mini | | GPT 4o | |
|---|---|---|---|---|---|---|
| **Ablations** | **TabFact** | **WikiTQ** | **TabFact** | **WikiTQ** | **TabFact** | **WikiTQ** |
| TableTextGrad $_{HA}$ | 86.56 | 66.30 | 86.35 | 62.89 | 88.42 | 73.02 |
| TableTextGrad $_{SA}$ | **87.05** | **70.58** | **86.62** | **64.14** | **88.75** | **75.10** |
| TableTextGrad $_{HL}$ | 86.76 | 66.66 | 85.11 | 60.29 | 87.12 | 72.96 |
| TableTextGrad $_{SL}$ | 86.62 | 68.58 | 84.86 | 61.20 | 88.20 | 73.24 |

In nearly all cases, tuning all prompts yields better performance compared to tuning only the last prompt. This suggests that fine-tuning the entire prompt chain allows the model to better optimize reasoning across all steps, not just the final output generation.

Tuning all prompts also consistently leads to superior or equal results across both datasets, regardless of the underlying model. This reinforces the importance of maintaining flexibility throughout the entire reasoning pipeline, as each prompt step contributes to more accurate responses, particularly in complex tasks such as WikiTQ. While last prompt tuned does not outperform full-prompt tuning, its competitive performance highlights the efficiency of tuning just the final step. For instance, with GPT 4.0 on TabFact, TableTextGrad $_{HL}$ achieves 87.12%, which is only slightly lower than the 88.75% of best-performing TableTextGrad $_{SA}$. This shows that, in resource-constrained

environments, tuning only the final prompt could offer a more efficient alternative with minimal performance trade-offs.

The performance gap between tuning all prompts and tuning the last prompt is slightly more pronounced in smaller models (e.g., GPT 4o mini), where full-prompt tuning tends to offer a greater boost in performance. This indicates that larger models like GPT 4o are more robust to freezing earlier prompts, likely because they possess stronger generalization capabilities.

## 4.2 Tuning All Prompts Vs Tuning Final Prompt

Similar to the common practice of fine-tuning only the last layer of a deep learning model, it is reasonable to hypothesize that fine-tuning just the final query prompt in the table QA pipeline could yield competitive results while reducing the computational cost. In this ablation, we explore the impact of fine-tuning only the final query table QA prompt while keeping all prior prompts in the reasoning chain frozen. The rationale behind this approach is that the earlier prompts are likely responsible for general task understanding and contextual reasoning, while the final prompt directly governs the model's response generation.

This ablation helps isolate the contributions of the final prompt in guiding table-based question answering, as well as assessing the role of prior prompts in contributing to overall system performance. If fine-tuning the last prompt yields performance close to full-prompt tuning, this approach could provide a significant efficiency advantage, reducing the number of parameters that require updating during training and consequently lowering memory and compute requirements. The results in Table 3 show that while tuning the final prompt alone achieves reasonable performance, it does not match the results of tuning the entire set of prompts. This suggests that earlier prompts play an integral role in step-by-step reasoning over table data, and their fixed nature might hinder the model's ability to fully optimize reasoning paths. However, the final prompt fine-tuning still offers a computationally efficient alternative, especially in scenarios with limited resources or when rapid deployment is required.

## 4.3 Effect of Table Length on Performance

We investigate the effect of lengths of tables on performance.

Table 4: Results on different table lengths. Small Tables are those where the sum of all the tokens of the table are <33 percentile. Medium are those >33 percentile and <67 percentile. Large Tables are those >67 percentile. We choose to show the results of the best-performing version of Table-TextGrad $_{SA}$.

| Table Lengths | Llama 3.1 70B | | GPT 4o mini | | GPT 4o | |
|---|---|---|---|---|---|---|
| | TabFact | WikiTQ | TabFact | WikiTQ | TabFact | WikiTQ |
| Small Tables | 91.12 | 81.81 | 87.50 | 66.14 | 92.52 | 83.96 |
| Medium Tables | 87.16 | 70.29 | 85.98 | 64.89 | 88.46 | 72.38 |
| Large Tables | 86.54 | 62.52 | 84.40 | 62.29 | 85.32 | 69.72 |

From Table 4, across all models and datasets, performance generally decreases as the table size increases. For example, with GPT 4.0 on WikiTQ, small tables yield an accuracy of 83.96, highlighting the increased difficulty in reasoning over larger tables where more tokens must be processed and contextualized. The highest performance is consistently seen on small tables across models and tasks. For instance, GPT 4o achieves 92.52% accuracy on TabFact and 83.96% on WikiTQ, which are the highest results for each dataset. This suggests that when the input is more concise, Table-TextGrad can reason more effectively, likely due to the reduced complexity and need for processing less information. As expected, large tables lead to the lowest performance. The increase in token count likely overwhelms the model's ability to capture relevant information efficiently, especially when complex reasoning is required. Smaller models like GPT 4.0 mini seem to have a lower ceiling, the 4% difference between small tables and large tables is small compared to larger models like GPT 4o, which drops from 83.96% to 69.72%. This indicates a higher sensitivity to the input length for more powerful LLMs. These results follow the trend of other models, such as Chain-of-Table and Dater.

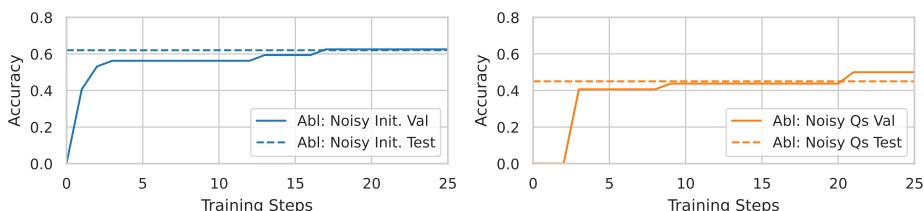

Figure 3: 2 experiments showing the effectiveness of TableTextGrad on noisy inputs. The experiment on the left starts with a poorly initialized final query prompt. The experiment on the right demonstrates TableTextGrad's ability to deal with noisy/irrelevant questions.

### 4.4 ROBUSTNESS TO POOR PROMPT INITILIZATIONS

In this section, we investigate a worst-case scenario where the initialized final query prompt is very poorly initialized. We perform experiments on a 200-sample subset of WikiTQ (100 for training, 100 for testing). The final prompt will be initialized as the following: `Here is a table and a question.  Return "I don't know".` (The usual prompt is shown in App. A.9.6) We also remove the ICL examples for the final query, so that the model has no information to work with, and has to learn how to answer the question from the training data starting from scratch. We use TableTextGrad $_{SL}$ to keep the maximum amount of information from the tables and only tune the final query prompt. From Figure 3, we see that TableTextGrad is able to achieve a respectable accuracy of 0.6 starting essentially from scratch. This highlights the power of TableTextGrad as well as the need for good initialization. The final prompt is in Appendix A.8.1.

### 4.5 ROBUSTNESS TO IRRELEVANT QUESTIONS

To test how our model performs a more difficult task with noisy input, we investigate a scenario where irrelevant information is added to questions to simulate an imperfect scenario. To do this, we add 4 randomly sampled questions from other tables so that the Agent has to identify the relevant question as well as answer it. Such a task would usually require significant methodology changes to address, but with TableTextGrad, the training step can automatically learn to parse out relevant information. For similar reasons as the previous experiment, we utilize TableTextGrad $_{SL}$. The results in Figure 3 demonstrate that TableTextGrad is indeed able to learn how to select and return the correct answer, at least 40% of the time. This simple experiment demonstrates the flexibility and usefulness of automatically tunable prompting pipelines. The final prompt is in Appendix A.8.2/

### 4.6 TRAINING PERFORMANCE MIRRORS ML TRAINING CURVES

This corresponds to the number of batches in $\mathcal{D}_{train}$ in Algorithm 2. For best performance, we run as many iterations as feasible with as many validation data points as possible. In our case, we run 32 iterations at 100 validation data points, sampled randomly for fairness. Note that we only chose a smaller number of validation datapoints since we have to run each one num_train_iterations times, which begins to become expensive. Each batch in the training set consists of 4 data points at each iteration. We found that batch size was relatively robust. See Appendix for more details.

Across all models and datasets, validation accuracy rapidly increases within the first few training steps (often before 10 steps) and then plateaus. This indicates that TableTextGrad quickly converges to a high level of accuracy during training. In general, the test performance aligns closely with the validation accuracy, suggesting that the small validation set is reasonably representative of the test set. This demonstrates that the model generalizes well from the validation set to the test set across different configurations. Larger models such as GPT 4o and LLaMA 3.1 70B tend to achieve higher test and validation accuracy compared to the smaller GPT 4o mini across both datasets. For instance, GPT 4o reaches near-perfect validation and test scores in both TabFact and WikiTQ, whereas GPT 4o mini shows a more gradual rise and slightly lower final performance. Both models generally perform better on TabFact compared to WikiTQ. This is evident from the higher plateaus

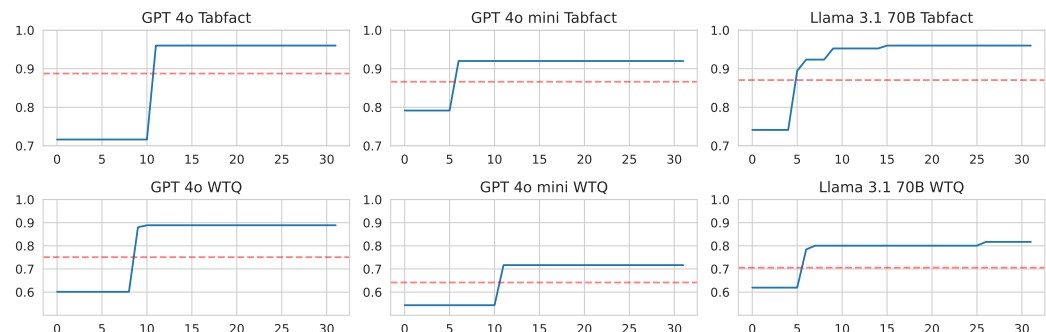

Figure 4: Validation accuracy of TableTextGrad on both the TabFact (top row) and WikiTQ (bottom row) datasets, with three different models: GPT 4o, GPT 4o mini, and LLaMA 3.1 70B. Each plot presents the validation performance (blue line) over the course of 32 training steps, and the test performance (red dashed line) is shown for comparison.

reached in validation accuracy for TabFact across all models. This trend likely reflects the additional complexity of WikiTQ, which requires more advanced reasoning over tabular data.

### 4.7 EFFICIENCY ANALYSIS

The efficiency of TableTextGrad is an important factor in its overall utility, especially compared to other table understanding approaches. Building on the relatively lightweight requirements of Chain of Table backbone, TableTextGrad's gradient-based refinement process incurs some additional computational costs. Specifically, the efficiency is driven by the fact that each gradient step requires only $O(10 \times$ number of training sample $\times$ number of validation points), where the maximum length of the table reasoning pipeline is 5, and each step in the pipeline outputs a response that also has to be backpropagated through. This means that the computational overhead scales with the size of the training set $\times$ validation set. Still, this is entirely manageable even for larger datasets, as seen in Section 4.6. We see that TableTextGrad converges closer to the beginning, potentially allowing for smaller amounts of training data. Given that many table understanding methods require more resource-intensive operations, such as full model finetuning or multiple self-consistency runs as in Dater, we argue that TableTextGrad 's approach is worth it to reduce the work of manual prompt optimization. A further discussion is shown in App. A.2.

### 5 CONCLUSION

In conclusion, table understanding presents a unique challenge, requiring both the comprehension of free-form questions and precise reasoning over semi-structured data. While recent prompting-based approaches leveraging Chain-of-Thought reasoning and function calls have shown promise without fine-tuning, the difficulty of designing effective initial prompts remains a critical barrier. Our proposed TableTextGrad framework introduces a novel extension of TextGrad principles to this domain, addressing the inherent complexity of conditional branching prompt pipelines. TableTextGrad not only demonstrates state-of-the-art performance on WikiTableQA, TabFact, and FeTaQA benchmarks but also proves to be robust and adaptable. Through experiments with poor prompt initialization and noisy questions, we illustrate its ability to recover and optimize performance under challenging conditions, showcasing its resilience compared to static, manually designed prompts. Moreover, experiments on prompt initialization robustness and robustness to noisy questions demonstrate the framework's flexibility, highlighting its potential for broader applications in table reasoning and beyond.

## LIMITATIONS

In its current form, TableTextGrad focuses on optimizing reasoning and prompt refinement for standard table reasoning tasks within the token limit constraints of large language models (LLMs). While the framework demonstrates state-of-the-art results on WikiTableQA and TabFact, handling very large tables presents a challenge due to the inherent length limitations of LLMs. These constraints can affect the efficiency of reasoning over tables with extensive rows and columns, where memory and attention span become critical bottlenecks.

To address this, TableTextGrad can be augmented with approaches such as TableRAG: Million-Token Table Understanding with Language Models or Tree-of-Table: Unleashing the Power of LLMs for Enhanced Large-Scale Table Understanding. Both techniques enable more scalable table understanding by partitioning or hierarchically structuring the table data to fit within the token constraints while maintaining semantic coherence. TableRAG Chen et al. (2024) introduces a retrieval-augmented mechanism, breaking large tables into smaller, manageable chunks and retrieving only the most relevant pieces for reasoning. Similarly, Tree-of-Table Ji et al. (2024) leverages a hierarchical attention mechanism that processes large-scale tables in a tree-like structure, enabling reasoning across expansive data while staying within the model's operational limits.

Integrating these methods with TableTextGrad would allow our framework to extend its applicability to large-scale tables, leveraging its iterative optimization capabilities on partitioned or hierarchically processed data. This combination not only addresses the token length limitations but also preserves the core advantages of TableTextGrad, such as its automated refinement of reasoning paths and robustness to noisy or poor initial prompts. We recognize this as a promising direction for future work, extending the utility of TableTextGrad to more complex and large-scale table reasoning scenarios.

Further future work could involve extending TableTextGrad to hierarchical table structures such as those found in HiTab Cheng et al. (2021). Hierarchical tables present unique challenges compared to flat tables, as reasoning often involves navigating nested relationships between rows and columns. Although this is currently not in scope with our existing work of flat tables, adapting our TableTextGrad could broaden its applicability to more complex and realistic real-world tabular datasets.

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

# Contents

# A APPENDIX

## A.1 ETHICS STATEMENT

This work on TableTextGrad was conducted using publicly available datasets, including WikiTable-Questions (WikiTQ) and TabFact, which are widely recognized benchmarks in the domain of tabular data understanding and reasoning. These datasets are accessible to the research community, ensuring that all evaluations and model training can be reproduced by other researchers under similar conditions. The use of publicly available data ensures transparency in evaluation and aligns with ethical practices of data usage and sharing within the machine learning community.

However, it is important to acknowledge that GPT models used in this work are proprietary and closed-source. The reliance on closed-source models poses some potential ethical challenges related to transparency, reproducibility, and equity of access. Researchers and practitioners outside of organizations with privileged access to GPT may find it difficult to replicate results or apply the model in their own work due to these restrictions. This limitation may hinder the open progress of scientific research and could create a barrier between institutions with access to proprietary models and those without, thereby limiting equitable advancements in the field. In contrast, LLaMA 3.1, which is used in this study, is an open-source model, enabling a wider range of researchers to replicate and extend the findings of this work. Open-source alternatives like LLaMA 3.1 help foster inclusivity and collaboration in machine learning research by lowering the barrier to entry for institutions and researchers globally.

### A.1.1 HUMAN IMPACT

The ability of TableTextGrad to improve the understanding and reasoning over tabular data holds significant potential for positive human impact. Tabular data is foundational in many domains,

including healthcare, finance, public policy, and scientific research. By enhancing the capabilities of models to analyze and reason over this type of data, TableTextGrad could improve decision-making processes across these fields. For example, in healthcare, better analysis of patient data could lead to improved diagnostic insights, while in finance, enhanced table understanding could streamline data-driven strategies and compliance efforts. This advancement can drive increased efficiency, better resource allocation, and more informed outcomes.

However, it is also important to recognize that the deployment of powerful AI models like Table-TextGrad must be approached with caution. The potential for automated systems to be used in decision-making processes could introduce risks if these systems are used without proper oversight. For example, inaccuracies in table interpretation or over-reliance on AI-generated insights could lead to misinformed conclusions, particularly in high-stakes areas such as healthcare or legal domains. Ensuring that TableTextGrad is deployed in a way that augments, rather than replaces, human judgment is critical for mitigating these risks. For example, prompt corrections should still be double-checked by a human for validity, to reduce the risk of hallucination.

## A.2 Cost Continued

Table 5: Table of cost of prompting baselines as well as TableTextGrad. TableTextGrad $_A$ indicates full prompt pipeline tuning and TableTextGrad $_L$ indicates only tuning the final query prompt.

| Method | Training Cost | # Inference Prompts |
|---|---|---|
| Binder | Manual Tuning | 50 |
| Dater | Manual Tuning | 100 |
| CHAIN-OF-TABLE | Manual Tuning | $\leq 25$ |
| TableTextGrad $_A$ | $\leq 25 \times$ # training data $+ 10 \times$ # training steps | $\leq 25$ |
| TableTextGrad $_L$ | $\leq 25 \times$ # training data $+ 2 \times$ # training steps | $\leq 25$ |

Table 5 provides a comparison of the prompting costs associated with baseline methods and Table-TextGrad, focusing on training effort and inference efficiency. Traditional methods such as Binder, Dater, and Chain-of-Table rely heavily on manual prompt tuning, which involves substantial human effort and domain-specific expertise. In contrast, TableTextGrad introduces a more scalable and automated approach to prompting through its iterative optimization framework. Both variants of Table-TextGrad, denoted as TableTextGrad $A$ and TableTextGrad $L$, substantially reduce the dependency on manual tuning by leveraging automated textual gradient optimization during training. Specifically, the cost for TableTextGrad is parameterized by the number of training data instances and training steps, where each number may be tuned in practice. At inference time, TableTextGrad requires no more than 25 prompts, matching the efficiency of Chain-of-Table. Notably, TableTextGrad $L$ is particularly efficient, requiring as few as 2 training steps per training data instance, compared to TableTextGrad $A$, which scales linearly with 10 training steps.

## A.3 Results on FeTaQA

In this section, we investigate TableTextGrad's performance on FeTaQA Nan et al. (2022), a free-form table QA dataset.

Table 6: Results on FeTaQA

| | BLEU | ROUGE-1 | ROUGE-2 | ROUGE-L |
|---|---|---|---|---|
| End-to-End | 28.37 | 0.63 | 0.41 | 0.53 |
| Dater | 29.47 | 0.63 | 0.41 | 0.53 |
| CHAIN-OF-TABLE (Rerun) | 31.46 | 0.65 | 0.42 | 0.54 |
| TableTextGrad $_{HA}$ | 33.75 | 0.67 | 0.44 | 0.55 |
| TableTextGrad $_{SA}$ | 34.06 | 0.68 | 0.46 | 0.56 |

From Table 6, we see that while TableTextGrad achieves higher BLEU and ROUGE scores compared to baseline methods, it is important to note that these metrics primarily reflect token-level matching rather than true semantic understanding or reasoning capabilities. As such, higher scores do not necessarily indicate improved performance on complex reasoning tasks but rather better alignment in token matching with reference answers.

## A.4  RESULTS ON FETAQA ROW AND COLUMN IDENTIFICATION

We perform an ablation to test the adaptability of TableTextGrad to predict relevant rows and columns. Note that in this scenario, we directly use a one-step prediction, bypassing all previous row/column selection functions. We perform experiments on a subset of FeTaQA dataset, with 200 samples (100 training, 100 test) and 25 training steps.

Table 7: Results on FeTaQA Row and Column Identification

|          | ROUGE-1 | ROUGE-L |
|----------|---------|---------|
| Rows     | 0.78    | 0.72    |
| Columns  | 0.79    | 0.60    |
| Combined | 0.82    | 0.72    |

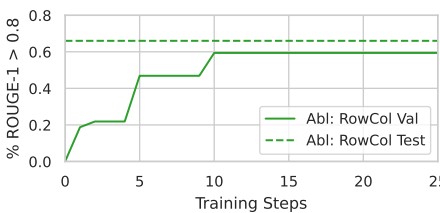

Figure 5: Training Curve of FeTaQA row / col prediction performance over 25 training steps.

Table 7 and Figure 5 present the results of TableTextGrad on row and column identification tasks for the FeTaQA dataset, which are crucial subtasks in table question answering (QA). These subtasks involve accurately aligning the question semantics with the relevant table rows and columns, enabling precise data retrieval for answer generation. TableTextGrad demonstrates robust performance on row and column identification, achieving a ROUGE-1 score of 0.78 and ROUGE-1 of 0.79 respectively, indicating its effectiveness. Notably, these results were achieved without requiring task-specific manual prompt tuning. The final learned prompt is the following:

```
 You are given a table.  The task is to return relevant rows and
columns based on the information in the table.
- Ensure all relevant rows and columns are explicitly included in
the response to capture the complete context of the question.
- Ensure the model identifies and uses consistent terminology and
capitalization for column names to prevent confusion.
- Ensure the model filters and focuses on only the relevant rows
and columns that directly pertain to the question.
- Ensure the response format is clear and structured, avoiding
unnecessary introductory phrases.
- Ensure the model verifies the accuracy of the data referenced
from the table before formulating the response.
- Ensure the model checks for potential ambiguities in the
question and clarifies them if necessary.
- Ensure the model provides a clear rationale for the inclusion of
specific rows and columns in its response.
- Ensure the final answer strictly follows the format:  "The
answer is:  row:  1,2,3.., column:  x, y, z ..."
```

## A.5  TABLE LENGTH VS PERFORMANCE ON WIKITQ

Table 8 demonstrates a fair comparison of the performance of the best-performing version of TableTextGrad on the test set. Other baseline results are taken from Wang et al. (2024). We see that TableTextGrad is able to obtain competitive performance against previous models.

Table 8: Accuracy of performance split by various table token lengths in WikiTQ.

|  | Small (<2k) | Medium (≥2k, <4k) | Large (>4k) |
|---|---|---|---|
| Binder | 56.54 | 26.13 | 6.41 |
| Dater | 62.50 | 42.34 | 34.62 |
| Chain-of-Table | 68.13 | 52.25 | 44.87 |
| TableTextGrad $_{SA}$ | 76.87 | 55.12 | 50.35 |

### A.6 REPRODUCIBILITY

All Llama 3.1 70B experiments were run on a server with 4 NVIDIA RTX A6000 GPUs (48GB VRAM), a AMD EPYC 7513 32-Core Processor, and 1000GB of RAM. The specific OpenAI GPT versions are gpt-4o-2024-05-13, and gpt-4o-mini-2024-07-18.

Code will be released after polishing and removing user-specific information.

### A.7 BATCH SIZE

Table 9: Results on different batch sizees.

| Batch Size | Llama 3.1 70B | | GPT 4o mini | |
|---|---|---|---|---|
|  | TabFact | WikiTQ | TabFact | WikiTQ |
| Batch Size 1 | 85.51 | 66.18 | 85.56 | 60.67 |
| Batch Size 4 | 87.05 | 70.58 | 86.62 | 64.14 |
| Batch Size 8 | 86.89 | 71.10 | 85.98 | 63.72 |

Table 9 demonstrates experiments on different batch sizes. We see that as long as the batch size is of reasonable, the performance is relatively consistent. Higher Batch sizes will require longer input lengths in the gradient step, so we limited our experiments to smaller sizes to avoid running into errors.

## A.8 EXPERIMENT PROMPTS

### A.8.1 ROBUSTNESS TO POOR PROMPT INITIALIZATIONS PROMPT

Here is a table and a question. Return the answer by extracting information specifically from *italicized* cells, as those have been determined to be relevant.

- Ensure the model summarizes key data points from the *italicized* cells succinctly, linking them directly to the question.

- Ensure the model formats the final answer strictly as "The answer is: AnswerName1, AnswerName2..." without additional commentary.

- Ensure the model avoids unnecessary phrases that do not contribute to the answer, streamlining the response for clarity.

- Ensure the model verifies the accuracy of the extracted data before formulating the final answer.

- Ensure the model checks for any missing or incomplete data in the *italicized* cells that may affect the answer.

- Ensure the model maintains a clear focus on the question being asked, prioritizing the identification of the relevant entity.

- Ensure the model provides a numerical representation of the answer when applicable, avoiding redundancy in the final answer.

A.8.2 ROBUSTNESS TO IRRELEVANT QUESTIONS PROMPT

```
 Here is the table to answer this question.  Please understand the
table and answer the question.

- Ensure the last line of the final answer is strictly "The answer
is:  AnswerName1, AnswerName2..." with no additional information
or context.

- Ensure only relevant *italicized* cells are referenced in the
answer, avoiding any unnecessary data.

- Ensure the final answer is concise and directly addresses the
question without extraneous elements.

- Ensure clarity by avoiding vague terms and providing complete
statements that directly address the question.

- Ensure the model identifies and prioritizes the most relevant
*italicized* cell(s) that directly answer the question.

- Ensure the model validates its answer against the table data
for accuracy before finalizing the response.

- Ensure the model explicitly identifies which parts of the
question are relevant to the provided table data.

- Ensure the model summarizes the relevant parts of the question
clearly, promoting coherence in the response.

- Ensure the model provides a brief justification for the selected
answer, explaining how it corresponds to the data in the table.
```

## A.9 EXAMPLE PROMPTS ORIGINAL VS TUNED

In this section, we demonstrate some examples of prompts that were turned by TableTextGrad. The full list of prompts will be released along with the code.

### A.9.1 GENERATE_PROMPT_FOR_NEXT_STEP

**Original**

```
 Choose the next operation in the function chain to answer the
question.  The output must start or add to the existing function
chain for the next operation.
```

**Tuned**

```
 Your goal is to construct a function chain that answers the given
question using the table data.  Choose the next operation from
the following options:  f_add_column() (to add a new column),
f_select_row() (to select specific rows), f_select_column() (to
select specific columns), f_group_column() (to group rows by a
column), f_sort_column() (to sort rows by a column), or <END> (to
finish the function chain).  Consider the context of the question
and the table data to choose the next operation.  Ensure that
each chosen operation logically follows from the previous steps
and contributes to answering the question.  Refer to the provided
examples to identify patterns in how operations are chosen based
on the question type.  Avoid operations that do not directly
contribute to answering the question or that might lead to dead
ends.  After choosing an operation, consider if it brings you
closer to answering the question.  If not, reconsider your choice.
```

### A.9.2 GROUP_COLUMN

**Original**

```
 To tell the statement is true or false, we can first use
f_group() to group the values in a column.  This count the number
of unique values in the column.
```

**Tuned**

```
 To answer the question, we can follow these steps:  1.  Identify
the relevant column(s) that contain the information needed.  2.
Perform the necessary operations such as filtering, counting,
or grouping the values in that column.  3.  Provide a clear and
concise explanation of the steps taken to arrive at the answer.
4.  Conclude with the column name used in the operation.

For example:  - If the question asks for a count, identify the
column to count, explain the counting process, and state the
column name.  - If the question requires filtering, identify the
column to filter, explain the filtering criteria, and state the
column name.  - If the question involves grouping, identify the
column to group by, explain the grouping process, and state the
column name.

Remember to handle edge cases, such as missing or incomplete data,
and verify the final answer by re-checking the data.
```

### A.9.3 SELECT_COLUMN

**Original**

> We can use f_col() to filter out useless columns in the table according to information in the statement and the table.

**Tuned**

> We can use `f_col()` to identify and return the relevant columns in the table by closely analyzing the information provided in the statement and the table. The function `f_col()` is used to encapsulate the relevant column names identified by the model. The output should be in the format: `f_col([column1, column2, ...])`.
>
> The model should link words and values in the statement to the corresponding columns in the table. Additionally, provide a detailed explanation for why these columns are relevant, considering both the keywords and the semantic meaning of the statement. Ensure that the explanation clearly links the statement to the columns.
>
> For example, if the statement is 'there are no cardiff wins that have a draw greater than 27,' the relevant columns would be 'cardiff win' and 'draw' because these terms are directly mentioned in the statement. For a more complex statement like 'in which three consecutive years was the record the same?', the relevant columns would be 'season' and 'record' because we need to check the values in these columns for consistency over three consecutive years.
>
> In cases where the statement does not directly link to any columns, provide an explanation of why no columns are relevant. If the statement links to multiple columns, provide an explanation of the links to each relevant column. Consider both the keywords and the semantic meaning of the statement. For example, if the statement implies a comparison or a trend, identify columns that can provide the necessary data for such an analysis.
>
> The output should include an explanation of the links between the statement and the columns, followed by the relevant column names in the format: `f_col([column1, column2, ...])`. Always list the relevant columns in the order they appear in the table. Ensure the explanation follows the format: 'The similar words in the statement link to columns: ... The column value in the statement links to columns: ... The semantic sentence in the statement links to columns: ...'
>
> By following these guidelines, the model can accurately identify and explain relevant columns in a table question answering task.

### A.9.4 SELECT_ROW

**Original**

> We can use f_row() to filter out useless rows in the table according to information in the statement and the table.

**Tuned**

> We can use `f_row()` to select relevant rows in the given table that directly support the explanation for the statement. For example, if row 3 is relevant, use `f_row([3])`. Please use `f_row([*])` to select all rows in the table. Always provide the row numbers in a list format, e.g., `f_row([3])` for a single row or `f_row([1, 2, 3])` for multiple rows. Your task is to provide an explanation for the answer and then specify the relevant row numbers using `f_row()`. Ensure your explanation is detailed and directly references specific data points in the table. Break down your reasoning step-by-step to ensure clarity. For example, if identifying the highest score, first state the criteria (e.g., highest score), then identify the relevant rows, and finally conclude with the row numbers. After providing your detailed explanation, clearly specify the row numbers at the end using `f_row()`. For example, 'The highest away team score is 23.11 (149), which is found in row 5. Therefore, the relevant row is 5. The answer is: f_row([5])`. Verify your explanation against the table data to ensure accuracy before specifying the row numbers.

### A.9.5 SORT_COLUMN

**Original**

> To answer the question, we can use f_sort() to sort the values in
> a column to get the order of the items.  The order can be "large
> to small" or "small to large".  The column to sort should have
> these data types:  1.  Numerical:  the numerical strings that can
> be used in sort 2.  DateType:  the strings that describe a date,
> such as year, month, day 3.  String:  other strings

**Tuned**

> To answer the question, we can use different operations based on the type of question.  The
> output must include a detailed explanation of the steps taken, the relevant column name, and
> the sort order if applicable.  Here are the steps and examples for each type of operation:
>
> 1.  **Sorting**:  – Use 'f_sort_by(column_name, order)' to sort the values in a column.  The
> order can be "large to small" or "small to large".  – Example:  To find the club in the last
> position, sort the "Position" column from large to small.
>
> 2.  **Filtering**:  – Use 'f_filter_by(column_name, condition)' to filter rows based on a
> condition.  – Example:  To find films with the language "kannada", filter the "language"
> column where the value is "kannada".
>
> 3.  **Counting**:  – Use 'f_count_rows(column_name, condition)' to count the number of rows
> that meet a specific condition.  – Example:  To count the number of films with the language
> "kannada", count the rows where the "language" column has the value "kannada".
>
> **Data Types and Operations**:  – **Numerical**:  Any column with numerical values (e.g.,
> integers, floats).  Operations:  sorting, counting.  – **DateType**:  Any column with
> date-related values (e.g., year, month, day).  Operations:  sorting, filtering.  – **String**:
> Any column with text values.  Operations:  filtering, counting.
>
> **Explanation Template**:  1.  Identify the type of question (sorting, filtering, counting,
> comparison).  2.  Determine the relevant column(s) and their data type(s).  3.  Choose the
> appropriate operation based on the data type.  4.  Provide a detailed explanation of the steps
> taken.  5.  Specify the column name and the sort order if applicable.
>
> **Handling Ambiguous Questions**:  – If the question is ambiguous or does not fit typical
> patterns, break down the question into smaller parts or ask for clarification by specifying
> the ambiguous part of the question.
>
> **Handling Comparisons**:  – For comparison questions, identify the relevant columns and
> compare the values directly or sort the relevant column to determine the highest or lowest
> value.
>
> **Error Handling and Edge Cases**:  – If the data contains missing values or inconsistent
> formats, first clean the data by removing or correcting these entries before performing the
> operations.
>
> **Common Pitfalls**:  – Avoid mixing up column names, misidentifying data types, or
> incorrectly applying operations.  Ensure the order of operations is logical (e.g., filter
> before sorting).
>
> By following these guidelines, we can effectively answer a wide range of table-related
> questions.  This structured approach ensures that the output includes a clear explanation,
> the relevant column name, and the sort order if applicable.

### A.9.6 FINAL_QUERY

**Original**

```
 Here is the table to answer this question.  Please understand the
table and answer the question - Ensure the last line of the final
answer is only "The answer is:  AnswerName1, AnswerName2..." form,
no other form.  - Ensure the final answer is a number or entity
names, as short as possible, without any explanation.
```

**Tuned**

```
 Here is the table to answer this question.  Please understand
the table and answer the question:  - Ensure you understand the
context of the table and the question before providing the final
answer.  - First, identify the relevant rows and columns.  Then,
calculate or extract the required information before formulating
the final answer.  - Ensure the final answer is only in the
form "The answer is:  AnswerName1, AnswerName2..." without any
additional text.  - Ensure the final answer is a number or entity
names, formatted as "The answer is:  AnswerName1, AnswerName2...",
without any additional explanation.  - If the data is ambiguous,
make a reasonable assumption, document it internally, and ensure
the final answer is consistent with this assumption.  - Verify the
extracted information against the table data before providing the
final answer.  - If uncertain, verify the extracted information
against the table data and provide the best possible answer in
the required format without indicating uncertainty.  - After
formulating the final answer, perform a post-processing step to
replace any en dashes with hyphens and remove any extra spaces
or special characters.  - Always provide the final answer in the
format "The answer is:  AnswerName1, AnswerName2..." without any
additional text or context.
```

