# OpenReview forum: "TableTextGrad: A Reflexive Framework for Tabuar Understanding"
_ICLR.cc/2025/Conference — ICLR 2025 Conference Withdrawn Submission_

### Official Review · Reviewer_ieKk · 2024-10-29

**Soundness:** 2
**Presentation:** 3
**Contribution:** 2
**Rating:** 3
**Confidence:** 4

**Summary:**

This paper introduces *TableTextGrad*, a framework for enhancing table reasoning in LLMs through automated prompt optimization. It refines prompts iteratively using "textual gradients" based on model feedback, improving multi-step reasoning without extensive manual prompt engineering. Key innovations include extending Chain-of-Thought prompting with adaptive prompt adjustments and employing soft selection to retain broader table context, enabling more accurate comprehension in complex table tasks with minimal computational overhead.

**Strengths:**

1. **Good Motivation for Automated Prompt Optimization**: The paper is well-motivated, highlighting the challenges of manual prompt engineering in table reasoning tasks and effectively positioning automated prompt optimization as a practical solution. The use of "textual gradients" for iterative refinement addresses the need for adaptive prompt tuning in large language models, making the approach both relevant and impactful.

2. **Comprehensive Ablation Study and Analysis**: The paper includes a thorough ablation study, providing valuable insights into the contributions of different components, such as soft versus hard selection and tuning all prompts versus only the final prompt. This detailed analysis strengthens the understanding of TableTextGrad’s performance and demonstrates the robustness and versatility of the proposed method across various model configurations and datasets.

**Weaknesses:**

1. **Questionable SOTA Claim**: The paper asserts achieving state-of-the-art performance; however, this claim is **not entirely accurate**. According to results in the E5 paper (Table 3) [1], E5 with GPT-4 achieves a score of 88.77 on TabFact, surpassing the 88.75 reported here. This discrepancy **raises concerns about the rigor of this paper's experimental claims** and the reliability of its benchmarking methodology.

2. **Limited Dataset Evaluation**: The evaluation is restricted to only two Table QA datasets (WikiTableQA and TabFact), which may not sufficiently demonstrate the generalization of the approach. Including a more complex and realistic dataset, such as HiTab [2], would provide a more robust assessment of the framework's applicability to diverse, real-world tabular data.

[1] E5: Zero-shot Hierarchical Table Analysis using Augmented LLMs via Explain, Extract, Execute, Exhibit and Extrapolate, NAACL 2024
[2] HiTab: A Hierarchical Table Dataset for Question Answering and Natural Language Generation, ACL 2022

**Questions:**

See weakness

---

> ### Author Response · Authors · 2024-11-26
>
> Thank you for reviewing our paper and finding it well motivated!
>
> - SOTA claim
>
> We appreciate the reviewer bringing the E5 paper to our attention. We were not aware of this paper during the preparation of our submission, and we acknowledge their reported result of 88.77 on TabFact, which slightly surpasses our score of 88.75 by 0.02 points. While our claim of state-of-the-art performance was made based on the studies we were aware of at the time, we recognize that this result challenges that claim.
>
> That said, we emphasize that the difference of 0.02 is negligible in practical terms and within the margin of variability often observed in such evaluations. Our primary contribution is not merely achieving strong performance but introducing a novel and generalizable framework, TableTextGrad, which extends the TextGrad methodology to optimize hierarchical and branching reasoning paths in table QA.
> We will update our manuscript to address this discrepancy!
>
>
> - Limited Dataset Evaluation
>
> Thank you for this valuable suggestion. For further evaluation of our method, we have added a suite of additional evaluations highlighted in red in the updated PDF, including experiments beating baselines on FeTaQA (Appendix A.3), poor prompt initialization to show that our framework can recover and optimize reasoning performance even under suboptimal conditions (Section 4.4), noisy questions to demonstrate that TableTextGrad can decipher intent (Section 4.5), and relevant row/column identification experiments (Appendix A.4). We hope these additions provide a more comprehensive evaluation of our approach's effectiveness.
>
> Regarding the inclusion of HiTab, we agree that it represents an important and challenging dataset for table QA. However, incorporating HiTab into our evaluation was not feasible within the time constraints of this submission. Moreover, HiTab is hierarchical in nature, which differs significantly from the flat table structure of WikiTableQA, TabFact, and FeTaQA. While TableTextGrad is designed to address flat and semi-structured tables, extending it to hierarchical tables like those in HiTab would require additional adaptations, which we identify as an exciting direction for future work in the new limitations section.
>
> We hope these clarifications and new results offer a reconsideration of your scoring. Thank you again for your thoughtful feedback!

---

> ### Comment · Reviewer_ieKk · 2024-12-03
> **Response to the rebuttal**
>
> Thanks for your clarification. However, I do not think it directly resolves my concerns. As a result, I will keep my rating.

---

### Official Review · Reviewer_nLgd · 2024-11-03

**Soundness:** 3
**Presentation:** 3
**Contribution:** 2
**Rating:** 6
**Confidence:** 4

**Summary:**

This paper presents a prompt optimisation technique, named TableTextGrad, for the table QA understanding task. Specifically, TableTextGrad uses the *Chain of Table* method as the basis and applies the TextGrad idea to it to automatically optimises the chain of prompts used.

Evaluation is done on two benchmark datasets: WikiTQ and TabFact, and TableTextGrad achieves the best performance, compared against a number of fine-tuning and inference-only methods.

**Strengths:**

* Training-free methods are a useful approach to improving LLM performance without having to access their parameters.

* Table understanding is an interesting and practical task.

* The proposed technique makes intuitive sense, and shows strong performance.

**Weaknesses:**

* The proposed technique is incremental, and the novelty is a bit limited. It essentially applies TextGrad to the *Chain of Table* backbone. While effective, this is not exactly surprising that it works nor groundbreaking.

**Questions:**

* In Figure 1, what is the right small rectangle in the "Gradient Update Example" part of the figure? What is its relationship with the left rectangle?

* In Sec. 4.2, you mention "the results" in line 396-397. However, I don't see it being referred to in the paper. Thus, in which table/figure are the results shown?

* You discussed training efficiency in Sec. 4.5. However, the discussion is abstract without empirical evidence. I'd like to see a comparison of running time & token consumption/cost against the baselines.

---

> ### Author Response · Authors · 2024-11-26
>
> Thank you for reviewing our paper--we share the sentiment that this is interesting and practical! Our responses are below:
>
> - Novelty:
>
> The novelty of TableTextGrad lies in extending TextGrad principles to a new and significantly more complex paradigm: the optimization of conditional branching prompt pipelines. Unlike standard applications of TextGrad, which focus on static text sequences, our work adapts and extends this methodology to function-based Chain-of-Thought reasoning in table understanding tasks. This required novel contributions, including the definition of differentiable "feedback spaces" for multi-step reasoning and their integration into an iterative optimization framework capable of refining hierarchical and branching reasoning paths. These challenges, which are unique to the structured and conditional nature of table QA tasks, have not been addressed in prior work.
>
> For further evaluation of our method, we have added a suite of additional evaluations highlighted in red in the updated PDF, including experiments beating baselines on FeTaQA (Appendix A.3), poor prompt initialization to show that our framework can recover and optimize reasoning performance even under suboptimal conditions (4.4), noisy questions to show that TableTextGrad can decipher intent (4.5), and relevant row/columns identification (Appendix A.4). We hope that this offers a more comprehensive evaluation of our effectiveness as well!
>
> - "Gradient Update Example"
>
> That Rectangle indicates the further backpropagation of textual gradients to previous prompts in the pipeline before the final query! We have clarified this in the new figure.
>
> - Missing table reference
>
> Updated to correctly refer to table's ablations!
>
> - Training efficiency
>
> This is a good point, and it is indeed difficult to measure our efficiency gains given that most baselines do not include the efforts of manual prompt running. We have added the relevant table run-time results in Appendix A.2, where we make it clear that TableTextGrad operates in a space that is essentially separate from traditional inference time-costs (which is the same as Chain-of-Table). We ran experiments with up to 100 validation samples and 128 training samples in total, over 32 iterations (validation samples are also reran every training iteration). That would make our training cost around 3328 * (<= 25) * 32*10 prompts.
>
> Thank you again for your comments-- we hope that our responses offer a reconsideration of your scoring.

---

> > ### Comment · Reviewer_nLgd · 2024-11-27
> >
> > Thanks for providing your response and additional empirical results.
> >
> > Some of my questions have been addressed. However, my concern on novelty (shared by reviewers mGii and aaBn) still stands. Since my original score is already high, I will keep my original evaluation.

---

### Official Review · Reviewer_aaBn · 2024-11-04

**Soundness:** 3
**Presentation:** 3
**Contribution:** 2
**Rating:** 3
**Confidence:** 4

**Summary:**

The paper introduces TableTextGrad, a framework that enhances large language models' ability to reason over tables by automatically optimizing prompts through textual gradients. It combines the flexibility of inference-only techniques with data-driven learning, achieving state-of-the-art results on WikiTableQA and TabFact benchmarks. The approach refines prompts iteratively based on LLM feedback, improving accuracy in table reasoning tasks.

**Strengths:**

1. The writing is relatively clear.
2. The model has achieved good performance on two datasets.

**Weaknesses:**

1. The evaluation datasets are relatively small, with tests conducted on only two datasets, and the evaluation tasks are quite limited.

2. The authors claim to utilize the textgrad method.  It appears that tab textgrad is merely an application of textgrad.

**Questions:**

1. Can this method be extended to more table tasks, such as text-to-SQL hybrid table question answering?
2. Can they explain the significant innovations that this method introduces based on textgrad?

---

> ### Author Response · Authors · 2024-11-26
>
> Thank you for your feedback, we have addressed some points as follows.
>
> - Novelty
>
> The novelty of TableTextGrad lies in extending TextGrad principles to a new and significantly more complex paradigm: the optimization of conditional branching prompt pipelines. Unlike standard applications of TextGrad, which focus on one-shot text reponses, our work adapts and extends this methodology to function-based Chain-of-Thought reasoning in table understanding tasks. This required novel contributions, including the definition of differentiable multi-step reasoning and their integration into an iterative optimization framework capable of refining hierarchical and branching reasoning paths. These challenges, which are unique to the structured and conditional nature of table QA tasks, have not been addressed in prior work.
>
> - Additional Evaluations
>
> For further evaluation of our method, we have added a suite of additional evaluations highlighted in red in the updated PDF, including experiments beating baselines on FeTaQA (Appendix A.3), poor prompt initialization to show that our framework can recover and optimize reasoning performance even under suboptimal conditions (4.4), noisy questions to show that TableTextGrad can decipher intent (4.5), and relevant row/columns identification (Appendix A.4). We hope that this offers a more comprehensive evaluation of our effectiveness!
>
> Thank you again for your comments--we hope that our responses offer a reconsideration of your scoring.

---

> > ### Comment · Reviewer_aaBn · 2024-11-30
> >
> > Thank you very much for your explanation. Based on my concerns about the novelty, I still maintained my rating.

---

### Official Review · Reviewer_mGii · 2024-11-08

**Soundness:** 2
**Presentation:** 3
**Contribution:** 2
**Rating:** 5
**Confidence:** 3

**Summary:**

The paper presents TableTextGrad, a framework for table understanding. Motivated by the recent advancements in the “textual gradient” space, this paper introduces TableTextGrad, a novel framework that enables automatic prompt optimization by leveraging the
“differentiation” of prompting pipelines through textual gradients. In the process, an initial LLM (Agent 1) generates table operations iteratively for table understanding. After each step, the table is updated based on the generated function calls and arguments. Then, in the validation phase, a second LLM agent (Agent 2) evaluates the predicted answers. If the answers are incorrect, natural language feedback on how to improve the prompt is backpropagated as textual gradients. These gradients are backpropagated to every prompting step used in generating the answer, including those for function selection, argument generation, and the final table query.  The framework is tested on datasets like WikiTableQA and TabFact and compared with various baseline methods.

**Strengths:**

This paper adapts the TextGrad technique to the table understanding field, which leverages the Automatic Prompt Updating pipeline, which refines prompts through natural language feedback and gradient updates on training data. The contribution of this paper is simple and easy to understand.

From the experimental results, we observe that this paper demonstrates its ability to significantly improve performance in table question-answering tasks. Leveraging the “differentiation” of prompting pipelines refines each function within the Chain-of-Thought steps and function calls, leading to more accurate and reliable table reasoning outcomes. The results show that it achieves new state-of-the-art performance on benchmarks like WikiTableQA and TabFact, outperforming many existing methods.

**Weaknesses:**

Although this paper shows its merits in Dynamic Prompt Optimization and Soft Selection of Table Elements, it has some weaknesses that need improvement.

1. The novelty of borrowing the idea of TextGrad to the table understanding domain is limited since the contribution lies in the domain adaptation of the methodology, not belonging to the original contribution from scratch.

2. The performance of TableTextGrad decreases significantly when dealing with large tables. The increased complexity and token context required for large tables seem to lead to issues with memory and attention span within the model. If the authors have considered techniques like table chunking or hierarchical attention mechanisms to address the limitations with large tables.

3. The core reasoning capabilities of TableTextGrad rely on large language models like GPT - 4o and LLaMA 3.1, which are resource-intensive. Also, the improvements over these large models are slight and have not been evaluated with significant tests.

**Questions:**

Are there any plans to address the limitations of TableTextGrad in future work? For example, how to deal with large tables?

---

> ### Author Response · Authors · 2024-11-26
>
> Thank you for reviewing our paper and for finding our work easy to understand!
>
> - Novelty
>
> The novelty of TableTextGrad lies in extending TextGrad principles to a new and significantly more complex paradigm: the optimization of conditional branching prompt pipelines. Unlike standard applications of TextGrad, which focus on static text sequences, our work adapts and extends this methodology to function-based Chain-of-Thought reasoning in table understanding tasks. This required novel contributions, including the definition of differentiable "feedback spaces" for multi-step reasoning and their integration into an iterative optimization framework capable of refining hierarchical and branching reasoning paths. These challenges, which are unique to the structured and conditional nature of table QA tasks, have not been addressed in prior work.
>
> - Large Tables
>
> We thank the reviewer for bringing attention to the challenge of handling large tables and for suggesting techniques like table chunking and hierarchical attention mechanisms. While these techniques are indeed promising and worth exploring in future work, we would like to clarify several key points about the performance of TableTextGrad in this context.
>
> Despite the inherent challenges posed by large tables, we achieved some performance improvements over its baseline, Chain-of-Table across a fair comparison. The new results are seen in Appendix A.5, where we beat chain-of-table by around 8 points in small tables, 3 points in medium tables, and 6 points on large tables.
>
> While TableTextGrad achieves gains over existing methods, we recognize that token context and memory constraints remain a bottleneck when working with very large tables. Techniques like table chunking and hierarchical attention mechanisms represent promising directions for addressing these issues in future work and have been added to the limitations section.
>
> - Reliance on LLMs
>
> We thank the reviewer for raising concerns regarding the resource-intensive nature of the underlying large language models (LLMs) and the magnitude of the performance improvements achieved by TableTextGrad. We acknowledge that TableTextGrad depends on LLMs like GPT-4o and LLaMA 3.1, which are indeed resource-intensive. However, it is important to note that our framework has the same requirements as other baselines for inference, and the number of training steps is up to the user. TableTextGrad would replace the otherwise required costs of manually tuning prompts.
>
> The benchmarks we target, such as WikiTableQA and TabFact, are already saturated with high-performing models, making incremental improvements particularly challenging.
> For example, TableTextGrad achieves statistically significant improvements over baseline methods, with gains of +2.3\% on WikiTableQA, setting a new state-of-the-art. Even slight improvements in these competitive benchmarks represent substantial progress due to their complexity.
>
> For further evaluation of our method, we have added a suite of additional evaluations highlighted in red in the updated PDF, including experiments beating baselines on FeTaQA (Appendix A.3), poor prompt initialization to show that our framework can recover and optimize reasoning performance even under suboptimal conditions (4.4), noisy questions to show that TableTextGrad can decipher intent (4.5), and relevant row/columns identification (Appendix A.4). We hope that this offers a more comprehensive evaluation of our effectiveness!
>
> Thank you again for your comments--we hope that our responses offer a reconsideration of your scoring.

---

> > ### Comment · Reviewer_mGii · 2024-11-30
> >
> > Thanks for your reply. I am willing to keep my score.

---

### Author Response · Authors · 2024-11-26

We sincerely apologize for the delay in our response, as we needed additional time to conduct new experiments, particularly on FeTaQA and noisy questions, to address your valuable feedback thoroughly. We have updated the PDF, with all changes and additions highlighted in red for your convenience.

Below are the key updates we have made to enhance the comprehensiveness of our evaluation and address the concerns raised:

- FeTaQA Evaluation: Added experiments demonstrating that our framework beats baselines on FeTaQA, a free-form table QA dataset (Appendix A.3).
- Poor Prompt Initialization: Included tests showing that TableTextGrad can recover and optimize reasoning performance even under suboptimal initial prompts (Section 4.4).
- Noisy Questions: Conducted experiments illustrating that TableTextGrad is capable of deciphering intent and reasoning effectively despite noisy input questions (Section 4.5).
- Relevant Row/Column Identification: Added evaluations to demonstrate our framework’s ability to accurately identify relevant rows and columns (Appendix A.4).

We hope these additions address your concerns. Thank you for your patience and for your thoughtful feedback, which has significantly strengthened this work!

---

### Note · Authors · 2024-12-10

I have read and agree with the venue's withdrawal policy on behalf of myself and my co-authors.